# Exploring the Effects of Intended Use on Targeting in Virtual Reality

Yuan Chen[1,3*]     Géry Casiez[3,2,1†]     Sylvain Malacria[3,1‡]     Edward Lank[1,3]

[1]University of Waterloo, Ontario, Canada
[2]Institut Universitaire de France, Paris, France
[3]Univ. Lille, Inria, CNRS, Centrale Lille, UMR 9189 CRIStAL, Lille, France

## ABSTRACT

Researchers have shown that distance and size are not the only factors that impact the target acquisition time in desktop interfaces, but that its intended use, whether it is selected, dragged, or otherwise manipulated, can also have a significant influence. However, despite the increasing popularity of virtual 3D environments, the intended use of targets in these contexts has never been investigated, in spite of the richer, multidimensional manipulations afforded by these environments. To better understand the effects of intended use on target acquisition in virtual environments, we present the results of a study examining five different manipulation tasks: targeting, dual-targeting, throwing, docking and reorienting. Our results demonstrate that the intended use of a target affects its acquisition time and, correspondingly, the movement towards the target. As these environments become more commonplace settings for work and play, our work provides valuable information on throughput, applicable to a wide range of tasks.

**Index Terms:** Human-centered computing—Virtual reality; Human-centered computing—Pointing

## 1 INTRODUCTION

Target acquisition is one of the most common actions performed in an interactive system regardless of whether the computing platform is a desktop computer [26, 37], a touch-based device [24], a large physical display [52] or an Augmented Reality environment [14].

Fitts's Law [37] is the most commonly used model to describe the movement time taken to acquire a target. Movement time (*MT*) for a one-dimensional pointing task is described by a linear function of the index of difficulty (*ID*) of the pointing task:

$$MT = a + b \cdot ID \tag{1}$$

where a, b are empirically determined regression coefficients and ID is a logarithmic term of target amplitude (A) and target width (W) [36]:

$$ID = log_2 \left( \frac{A}{W} + 1 \right) \tag{2}$$

However, target amplitude and width are not the only factors impacting acquisition performance [63]. One factor that may impact target acquisition time is the *intended use* of the target. By *intended use*, we refer to manipulations that a user intends to do or will do with the target they acquire. For example, in virtual environments, beyond simply acquiring an object, users can scale objects and move them around in the space [62], rotate objects to reveal the occluded view [16] or manipulate objects' motion to simulate physical phenomena [4]. In order to perform any of these intended uses of a target, we

*e-mail: y2238che@uwaterloo.ca
†e-mail: gery.casiez@univ-lille.fr
‡e-mail: sylvain.malacria@inria.fr

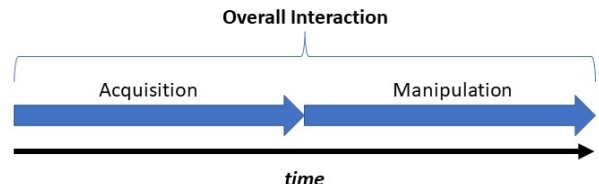

Figure 1: **The overall interaction with an object includes two sequential steps: acquiring the object – time characterized by Fitts' Law or a variant – and manipulating the object – time varies depending on the complexity and nature of the manipulation.**

must first acquire it. Acting on an object in an interface is thus a compound action comprised of acquiring it and performing the action or manipulation of it (See Figure 1) and the question we pose in this paper is whether the manipulation that we will perform on an object in a virtual environment, our intended use, impacts the performance of acquiring the target.

In 2D computer interfaces, research has shown that target acquisition time may vary depending on whether the user wishes to simply acquire a target (e.g. targeting), move it (e.g. dragging or docking), or throw it (e.g. flicking) – independent of the time taken by the subsequent task [38, 48]. It is always tempting to assume that results in 2D can be directly applied to 3D, particularly to Virtual Reality (VR), but we hesitate to make such assumption for several reasons. For instance, the controller in VR is an absolute input while the mouse is a relative input, leading to different acquisition and manipulation behaviours, and some tasks more frequent in 3D were not investigated in 2D (e.g. reorienting a target). Besides, to the best of our knowledge no similar analysis has been performed in the increasingly common context of direct target acquisition in VR. As we incorporate VR into both work [18, 35] and entertainment [51], understanding the impact of various independent variables on performance measures for VR-based selection tasks remains an area of active research interest (see [10] for a review).

To explore the impact of intended use on targeting in virtual environments, more precisely in the context of proximal interaction using the virtual hand metaphor, we examine five common manipulations in various VR systems, such as games [3], social [39] and educational applications [25] – TARGETING (selecting a target), DUALTARGETING (selecting one then another target), THROWING (acquiring and pushing a target), DOCKING (acquiring and placing a target) and REORIENTING (rotating a target about its axes) – and measure how long it takes to acquire the target to be manipulated. We find that simple targeting exhibits the fastest acquisition times for that target, reorienting the slowest acquisition times, and other intended target manipulations result in acquisition times between these extremes. Given the initial finding that the intended use of the target impacts time, we probe additional characteristics of movement toward a target with the goal of understanding how and why prior movement time is impacted by the subsequent intended use of the target. Movement profiles highlight both characteristics and differences in peak speed, time to peak speed, movement prior to

selection and selection speed across different intended uses.

This paper is organized as follows. After presenting the related work on target acquisition modelling in both 2D and 3D, we describe and explain the experiment setup and task design of our study. Then, we detail our analysis from three aspects: selection time, motion kinematics and Fitts' law modelling. Finally, we present our findings and discuss their implications.

In summary, this is the first work to explore the impact of intended use on targeting in VR and our contributions include:

- Investigating impacts of intended use on target acquisition in VR and summarizing characteristics of each use.

- Presenting design implications for interfaces and interactions in VR.

## 2 RELATED WORK

Fitts' Law [22, 37] is probably the most well-examined relationship in human-computer interaction (HCI) research. While Fitts' Law was originally formulated in terms of a 1D pointing task, researchers in HCI have long recognized that understanding the cost of target acquisition in graphical interfaces is useful, as it allows us to characterize the relative efficiency of different arrangements of interfaces. As a result, researchers have proposed a number of extensions to Fitts' Law to model 2D targeting [2], 3D targeting [54], gaze-based targeting [58], foot-based movement [28, 57], among others. Researchers have also explored modelling error in Fitts' Law [60], generalized Fitts' Law to incorporate steering tasks [1, 42], and leveraged the fundamental components of Fitts' Law to design a host of pointing facilitation techniques (see [6] for a review).

The goal of research into Fitts' Law is to understand and improve throughput [63] in the use of interactive systems. If we characterize the temporal cost and error rate of individual interactions (e.g. selecting a target, manipulating that target, keyboarding, homing, etc.), then the overall temporal cost and error rate of a compound interaction is the sum of these basic interactions [13, 19]. By improving the speed of target acquisition via new target acquisition techniques [6] and the speed of interactions via new interaction techniques [10], each of these subtasks becomes more efficient, increasing the overall efficiency of the interaction. Essentially, the assumption is that each individual interaction can be independently optimized. This is equally true in VR research: Bergström et al. [10], analyzing 20 years of VR research, note that studies with selection tasks measure the time a participant needs to select the next target (occasionally with additional measures of error and throughput for the selection task), while studies with manipulation tasks measure the time a participant manipulates virtual objects.

While, to the best of our knowledge, the independence of basic interactions has never been evaluated in VR, we have reason to believe that this assumption is questionable in traditional two-dimensional computer interfaces. Mandryk and Lough [38] examined how the *intended use* of a target impacts the time it takes to acquire the target. Mandryk and Lough note that, in real-world interfaces, the user acquires a target with a specific goal in mind. Perhaps they wish to click the target (targeting). Perhaps the target activates a secondary set of widgets and they need to then click on a second target (i.e. dual targeting). Perhaps they wish to move the target in some way, e.g. to re-position it imprecisely (flicking or dragging) or to re-position it precisely at a new location (docking). Mandryk and Lough found that, if the intended use was targeting or dual targeting, participants acquired the target significantly faster than if the intended use was flicking or dragging. They also noted a difference in acceleration and deceleration during the selection of the target to be manipulated: if the participants intended to flick or dock the target, then the selection movement toward that target exhibited a higher peak speed and a longer deceleration phase. Follow-on work by Ruiz and Lank [48] replicated these results and, via a more complete

analysis of movement profiles, analyzed their potential impact on kinematics-linked endpoint prediction [34]. While Ruiz and Lank note that the impact on kinematics-linked modelling was likely not a concern, in both cases results implied that the potential benefits of new interaction techniques observed during manipulation may not be realized if they result in a corresponding increase in acquisition time for the target to be manipulated.

There is also a significant possibility that virtual reality manipulations will differ from both real-world and two-dimensional interface manipulations. Considering real-world manipulations, the field of psychology has actively studied the act of reaching and grasping for many years [47]. Factors, including perception of the object to be grasped [23], the manipulations to be performed on the object [50], and tactile feedback during the act of grasping [30] impact both trajectories toward an object and the positioning and speed associated with the grasping of an object. However, there exists a disconnect between real world affordances and perceived affordances [43] and an absence of the physiological interactions between hand and object, which may impact behaviour. Furthermore, while it is always tempting to assume that previous results in 2D [38, 48] can be directly applied to 3D, especially to VR, past research indicates that this may not be true. As one example of this, Cockburn and Mckenzie [17] found that user performance deteriorated for a locate-and-point task when transforming from a 2D interface to a 3D interface. Furthermore, movement planning, whether in real world or in computer interfaces, requires trajectory planning [34, 47]; in immersive VR environments, depth has been found to greatly impact both perceived width and distance [5, 15], a factor that may significantly alter the trajectory, kinematics and the impact of intended use.

## 3 METHODOLOGY

We conducted a controlled experiment to investigate the effect of intended use of a target on the time taken to acquire it in VR. We focused on interaction at arms' length using a 1-to-1 mapping of physical controller movement to virtual controller movement, a direct 3D target acquisition technique common in VR [15, 45, 46]. Figure 2 depicts the timeline of a trial in this experiment.

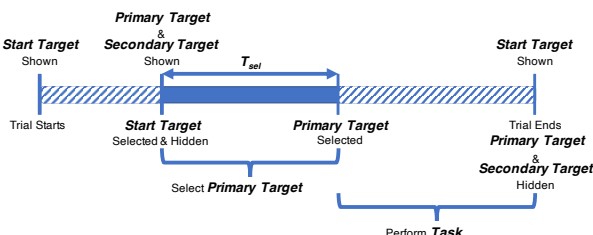

Figure 2: A timeline of a trial: A trial starts by showing the *Start Target*. Once it is successfully acquired, it vanishes and a *Primary Target* is shown, together with a *Secondary Target* if required by the *Task*. Participants acquire the *Primary Target* and perform the *Task*. *Start Target* is shown again when the trail ends.

### 3.1 Apparatus and Participants

The system was implemented in Unity 2020.3.7f1 with the Oculus Integration v29.0 and the study was conducted standalone on the Oculus Quest 2 (tracking frequency is 72 Hz) with Oculus left- and right-hand controllers.

In pilot studies we noted that the virtual controllers could occlude targets and their irregular shape caused problems of precision when selecting a target (because the exact selection location was unclear). To address this, we used a smaller controller 3D model (from GEAR VR), made the models translucent, and added a blue sphere (**cursor**) to the top side of the virtual controller indicating selection location, as shown in Fig. 3. Although the physical and virtual controllers

are not in the same shape, this design rarely affects how participants recognize them as participants only see virtual controllers rather than physical ones. In addition, participants focus on using **cursor**, rather than the model for interactions. Making the model semi-transparent also does not impact the selection performance of a virtual hand [32, 56]. Besides, **cursor** is useful for target selection in VR [7] and our design presents better visual tracking on it.

We also noted that, as participants became fatigued, they would sometimes switch hands for a brief period, interacting with their non-dominant hand which impacted performance. Therefore, alongside instructing participants to perform the study only with their dominant hand, we disabled the non-dominant hand controller at the outset of the study and invited participants to take breaks during the experiment if needed.

A total of 15 participants, aged from 22 to 33 ($M = 26.1, SD = 3.1$, 5 identified as women, all right-handed) participated in the study. 12 participants had experienced VR prior to the study; those participants primarily used VR for entertainment activities, such as playing games. The experiment took approximately 45 minutes and participants received $15 for their participation.

## 3.2 Interaction for Target Acquisition

Fig. 3 illustrates the visual interaction of the target acquisition. To acquire a target, participants place the cursor (1 cm width) partially inside a target and press the selection button (in our implementation, A on the right controller) to select that target. When any part of the cursor enters a target, the target is highlighted in translucent blue, indicating that it is now selectable. A successful selection turns the cursor green. If instead the button is pressed while the cursor is outside the target, the cursor turns red and an alert sound is played to inform participants of the erroneous selection. Participants are expected to correctly acquire each target, and they can proceed after a correct selection.

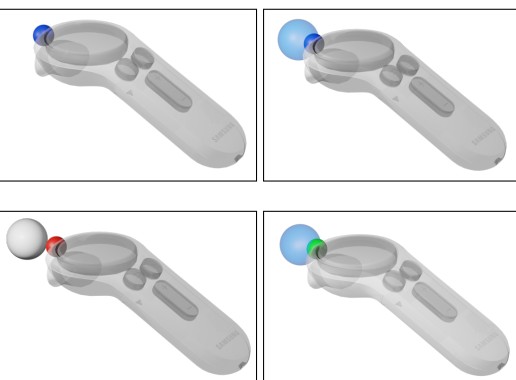

**Figure 3: Input controller: when *Cursor* enters a target, the target turns translucent blue. Incorrect selections turn *Cursor* into red while correct selections turn it into green.**

Given that the overall goal of this experiment was to measure how the intended use of a target impacts the time taken to acquire that target, the overall interaction requires acquisition of a target followed by a manipulation of that target, i.e. an intended use or *Task*. The interaction to start a *Task* proceeds as follows:

1. A participant acquires an initial target, the *Start Target*. To ensure that all targets in different tasks are noticeable and reachable within the field of view, the *Start Target* is positioned 35 cm in front of and 20 cm below the head.

2. When the participant correctly selects the *Start Target*, the *Start Target* vanishes and a *Primary Target* is displayed immediately.

He/She is asked to acquire and manipulate this *Primary Target*. Considering that the arms of healthy adults typically reach at least 60 cm from the torso [5, 15, 31, 33, 44], targets are placed within 60 cm of participants in a region surrounding the location of the *Start Target*.

3. Participants move the cursor from the *Start Target* to the *Primary Target* and perform a *Task* on the *Primary Target*.

By default, the *Start Target*'s position remains the same in the virtual world for all tasks and targets are anchored relatively to the *Start Target*'s position, which do not follow participants' movement. However, as participants could adjust their position during the experiment, we allow them to re-calibrate the *Start Target*'s position when they notice a shift in their position, before starting any new *Task*. In practice, the re-calibration was seldom used.

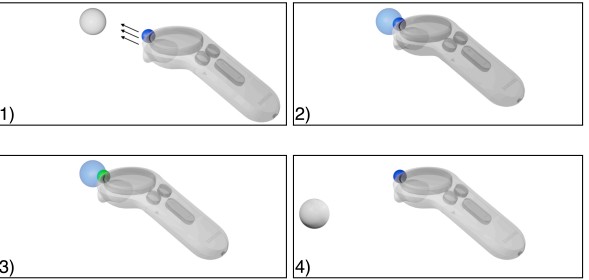

**Figure 4: Correctly acquiring *Start Target* to calibrate starting position and reveal the task (the example shows TARGETING).**

## 3.3 Independent Variables

Independent variables (IVs) in this study include the Index of Difficulty (*ID*) of the *Primary Target*, *Primary Direction*, *Secondary Direction* and *Task*.

*ID* of the acquisition action is computed using MacKenzie's formula [36] for the Amplitude **A** (distance between the *Start Target* and the *Primary Target*) of values 9 cm, 12.5 cm, or 16 cm, and the Width **W** of the *Primary Target* of values 3 cm or 5 cm). This yields six different IDs (1.49, 1.81, 2.00, 2.07, 2.37, and 2.66 bits).

*Primary Direction* **D** and *Secondary Direction* **D'** (Up, Down, Forward, Backward, Left or Right) represent the direction of movement from the *Start Target* to the *Primary Target* and the *Primary Target* to the *Secondary Target* respectively.

The rationale for these values is as follows. The *Primary Target* can be located in six basic directions from the *Start Target*, i.e. Up, Down, Forward, Backward, Left and Right. Given the position of the *Start Target*, six distinct directions, and the possible existence of sequential manipulation after acquiring the *Primary Target*, the maximum values of target amplitude and target width are set to 20 cm and 10 cm respectively to avoid a target appearing outside of the field of view or outside of the reachable workspace. These constraints motivate the above values for independent variables. With this configuration, participants observe targets that are close to them from a top view rather than a straight horizontal view, which helps to reduce the depth impact on the perceived width and addresses the occlusion issue between *Primary Target* and *Secondary Target*.

### 3.3.1 Task

The *Tasks* that participants were asked to complete were one of five manipulations of the primary target: TARGETING, DUALTAR-GETING, DOCKING, THROWING, or REORIENTING (illustrated in Fig. 5). Detailed description of individual tasks follows.

**TARGETING:** Correctly selecting the *Start Target* of width W reveals a white sphere (*Primary Target*) of width W located at the

amplitude A in direction D from the position of *Start Target*. Participants simply have to acquire the *Primary Target* by moving the controller to the target and selecting it.

**DUALTARGETING:** Correctly selecting the *Start Target* of width W reveals a white sphere (the *Primary Target*) and a red sphere (the *Secondary Target*) both of width W. The *Primary Target* is located in direction D with amplitude A from *Start Target* while *Secondary Target* is located in another direction $D'$ (different from previous and next trials) with amplitude 9 cm from the *Primary Target* (a distance that guarantees that both targets still remain within arms' reach for the participant, see Sect. 3.3). Participants must select these two spheres sequentially: first the *Primary Target*, and then the *Secondary Target*.

**DOCKING:** Correctly selecting the *Start Target* of width W reveals a white sphere (the *Primary Target*) of width W and amplitude A in direction D, and a semi-transparent sphere (the *Secondary Target*) 1.5 times larger than the *Primary Target* and located 9 cm away from the *Primary Target* in direction $D'$. The width of the *Secondary Target* reduces the required precision of the task, allowing participants to finish this task more easily. Participants are instructed to drag-and-release the *Primary Target* into *Secondary Target*.

**THROWING:** Correctly selecting the *Start Target* of width W reveals a white sphere (the *Primary Target*) of width W located at a position direction D from *Start Target* and at a distance of amplitude A, and a semi-transparent green wall (the *Secondary Target*) located 9 cm away from the *Primary Target* in direction $D'$. To reduce task difficulty (wall size being too small) and avoid visual distraction (wall size being too big), the size of this wall is set to 40 cm × 40 cm × 1 cm which is over 8 times larger than the size of *Primary Target* in width. Participants are instructed to *"throw"* the *Primary Target* towards the *Secondary Target* by releasing the pressed button. The released *Primary Target* then moves in the throwing direction.

**REORIENTING:** Correctly selecting the *Start Target* of width W reveals a white object (the *Primary Target*) and a red object (the *Secondary Target*) whose bodies are both spheres of width W. Different shapes (i.e. a capsule, a cylinder and a cube) are placed into both *Primary Target* and *Secondary Target* to indicate the orientation, which only serve as visual references and are not selectable. The *Primary Target* is located at amplitude A from *Start Target* in direction D and *Secondary Target* 9 cm away from *Primary Target* in direction $D'$. The *Secondary Target* has an orientation along its roll axis with a random angle in the range $(-\frac{\pi}{2}$ to $\frac{\pi}{2})$, values empirically obtained from pilot studies to reduce clutching and task difficulty. To move to the next trial, participants were asked to rotate the *Primary Target* such that the *Primary Target* has a similar orientation to the *Secondary Target*, that is when the angle difference in each axis is smaller than $\frac{\pi}{12}$. The colour of *Secondary Target* changes to green to inform participants of the correct orientation of the *Primary Target*.

Except TARGETING, all other tasks require a manipulation sub-task on either the *Primary Target* or the *Secondary Target*. For DUALTARGETING, a second selection is required on the *Secondary Target* and other tasks require a "press and hold" behaviour on the *Primary Target* for object manipulation. While behaviours in the manipulation sub-task vary across *Task*, the acquisition time of the *Primary Target* (delimited by the controller "button down" action on the primary target) is consistent across all tasks.

### 3.4 Procedure

Participants were recruited from our local university. To preserve social distancing requirements, the study was conducted remotely, and both the VR headset and controllers were sanitized before being delivered to participants.

Before written consent was obtained, participants were asked to read an information letter in which they were warned about potential

**(a)** TARGETING

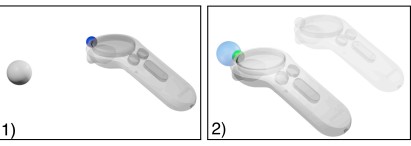

**(b)** DUALTARGETING

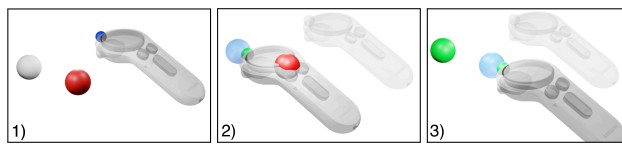

**(c)** DOCKING

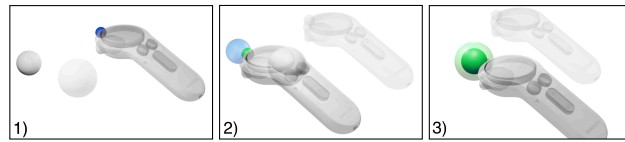

**(d)** THROWING

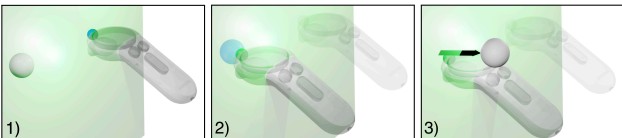

**(e)** REORIENTING

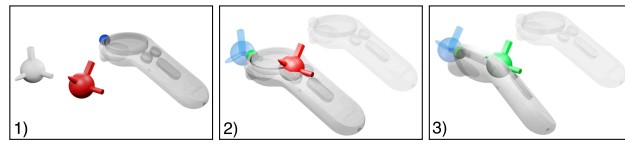

**Figure 5: Illustration of visualization, selection and manipulation actions in each task. 1)** *Primary Target (white) appears at the left of the vanished Start Target and Secondary Target appears at the back of the Primary Target.* **2)** *Correct selection on the Primary Target.* **3)** *Perform the manipulation actions on either Primary Target or Secondary Target based on the task.*

motion sickness as a result of wearing the head-mounted display device and that they could stop the experiment at any time without penalty if they felt uncomfortable or simply did not wish to continue. They were then asked to watch an instructional video, answer a demographics questionnaire (asking for their gender [53], age, handedness, and VR experience). They were then asked to sign the informed consent.

Participants started the experimental software by connecting with one of the researchers via video conferencing tools. The researcher verified informed consent, walked participants through disabling the non-dominant hand controller, and then guided the participant to start the experiment. In addition to researchers' verbal guidance, participants could also follow the instructions text in the system during the experiment, which presented detailed steps to guide them to control the system and finish each task.

As noted above, to evaluate the impact of intended use on target acquisition in VR, we represented different intended uses as a *Task* for the participant. The experiment consisted of repeated blocks of trials where participants had to select a target and complete a given *Task* on that target using the controller in their dominant hand.

Each trial, therefore, consisted of the following steps. First, the

participant had to acquire the *Start Target* to calibrate a starting position. Correctly selecting this target would hide it, start a countdown timer, and reveal a *Primary Target* that the participant was asked to acquire and manipulate to fulfill the *Task* (Section 3.3.1) for the given condition. A trial ended once the *Task* was completed successfully or if the countdown timer exceeded 15 seconds. The countdown timer was implemented to avoid excessive trial completion times, i.e., to limit study duration. The experimental system then moved to the next trial and revealed the *Start Target* again (see Fig. 4 & Fig. 2).

Participants were instructed to complete the trials as quickly and accurately as possible while keeping an error rate below 5% (error rate was displayed to the participant at the end of each block). In case of an error, the trial was appended to the end of the current block, and participants needed to complete all trials within a block successfully before moving on to the next block, thus ensuring the same number of correct trials for all conditions. While participants were not required to repeat a block if the error rate exceeded 5%, controlling the error rate is a common practice to balance speed and accuracy in pointing experiments [7]. Participants were allowed to take a break without a time limit between each block and after completing each *Task*.

## 3.5 Data Collection

All participant movement data was logged as a sequence of time-stamped, three-dimensional coordinates. The system also logged selection actions, whether the selection was an error and subsequent task manipulations.

Recall that the goal of this experiment is to measure the effect of different *Task*s (intended uses of the *Primary Target*) on the time taken to select the *Primary Target*. The dependent variables collected and logged by the system were selection time ($T_{sel}$) and errors. For all trials, we use button press down events on the *Start Target* and *Primary Target* to determine the beginning and end of the *Primary Target*'s selection movement. The selection time was the time interval from the button press action of the *Start Target* to the button press action on the *Primary Target*. The length of time to complete the *Task* was logged, but is immaterial to this experiment as we are only interested in the impact of *Task* on *Primary Target* selection time.

As noted above, the system also identified errors. We consider errors that occurred only while acquiring the *Primary Target*. Errors were classified as one of three error types:

- enter error — when participants pressed the button outside the *Primary Target* before having entered it.

- exit error — when participants pressed the button outside the *Primary Target* after having entered it at least once

- pass error — when the countdown timer reached 0 before the *Primary Target* was correctly selected.

The Error rate, displayed to participants and in our analysis, refers to the percentage of erroneous trials in a block over the total number of trials in the block (including any repeated trials). For example, if a participant failed once on every trial within a block and then succeeded on the second attempt, the error rate would be 50%.

## 3.6 Design Summary

We adopted a repeated-measures within-subjects design. We effectively looked at three independent variables (IVs): *Task* (TARGETING, DUALTARGETING, DOCKING, THROWING and RE-ORIENTING), *ID* (1.49, 1.81, 2.00, 2.07, 2.37, and 2.66 bits), and *Block* (1-4). The order of *Task* was counterbalanced across participants using a Latin square [59]. Note that for each *ID*, *Primary Direction* **D** and *Secondary Direction* **D'** were randomly ordered for

generalization. The combination between **D** and *ID* and that of **D** and **D'** were not controlled in our experiment. In summary, each participant completed 4 *Block*s × 6 *ID*s × 6 *Primary Direction*s × 5 (counterbalanced) *Task*s, i.e. 720 trials. This resulted in a data set containing 10 800 successful trials for our 15 participants.

## 4 RESULTS

We analyze our results in terms of error rate, *Primary Target* selection time ($T_{sel}$), motion kinematics and Fitts' Law modelling. Note that our focus is on $T_{sel}$, the time taken to move from the *Start Target* to the *Primary Target* given that our interest is on how *Task*s performed on the *Primary Target* impact the time to select it, $T_{sel}$.

### 4.1 Error Rate

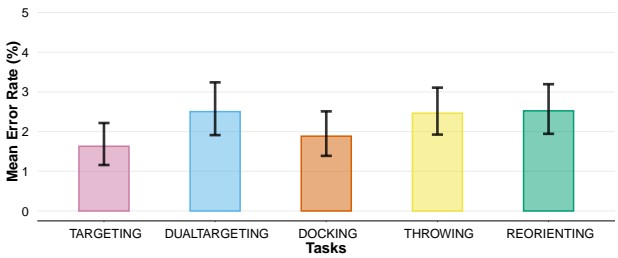

**Figure 6: Mean error rate of $T_{sel}$ for *Task*. Error bars are shown with 95% confidence intervals.**

Recall that, in order to complete a block, participants were required to successfully complete each trial. In the case of an error in selecting the *Primary Target*, participants would need to repeat the corresponding trial at the end of the Block. As a result, alongside the 10,800 correct trials, we collected an additional 289 erroneous trials, for a total of 11,089 trials. Among 289 trials, only 3 trials had pass error and participants did not report the time pressure during the study, implying that the timer did not push participants.

Given the non-normal distribution of error rate, a Friedman test was conducted for three independent variables (IVs): *Task*, *Block*, and *ID*. We found a significant effect of *Block* on error rate ($\chi^2_{Block}(3)$=15.46, p<0.005). However, pairwise Wilcoxon rank sum tests with Bonferroni corrections did not show significant differences between blocks. With all blocks, error rate (*M*=2.20%, *SD*=5.77) was below the 5% error rate threshold that we recommended our participants not to exceed. Error rate of each *Task* was shown in Fig. 6. The Friedman test did not reveal a significant effect of *Task* on error rate, but showed a significant effect of *ID* ($\chi^2_{Block}(3)$=12.92, p<0.05). However, pairwise comparisons did not reveal any significant differences across *ID*s.

### 4.2 Selection Time

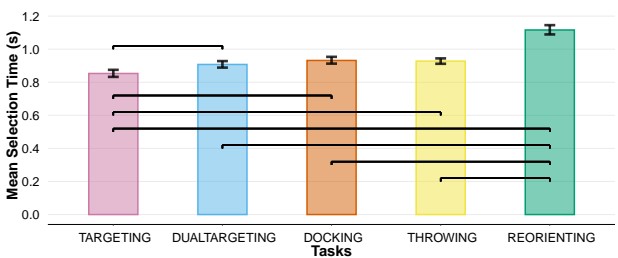

**Figure 7: Mean selection time of $T_{sel}$ for *Task*. Error bars are shown with 95% confidence intervals. The statistic significances evaluated by pairwise t-test are connected with lines (p<0.01).**

We aggregated non-erroneous trials and removed outliers by eliminating any trial whose selection time was more than three standard deviations from the mean, leaving 10,645 trials for analysis.

Given the non-normal distribution of the data, a Box-Cox transformation [11] was applied to selection time. When sphericity was violated using Mauchly's test, Greenhouse-Geisser correction to the DoFs was applied. When significant effects were found, pairwise t-tests with Bonferroni corrections were conducted for post-hoc analysis. Effect sizes were reported as partial eta squared ($\eta_p^2$).

We first conducted a two-way RM-ANOVA ($\alpha$=0.05) for selection time on *Block* and *Task* to test for a possible learning effect. We found a significant effect of *Block* ($F_{3,42}$=41.64, p<0.001, $\eta_p^2$=0.75). Pairwise comparisons revealed significant differences between Block 1 (M=1.03s) and the other three blocks (p<0.001, 2: 0.97s, 3: 0.94s & 4: 0.93s). We found a significant effect of *Task* ($F_{4,56}$=33.11, p<0.001, $\eta_p^2$=0.70) but we did not find a significant interaction effect between *Block* and *Task*. These results suggested a potential learning effect that did not differ between tasks; we thus removed the first block in our remaining analysis.

After removal of the first *Block*, we found a significant effect of *Task* ($F_{4,56}$=29.11, p<0.001, $\eta_p^2$=0.68) on $T_{sel}$. As shown in Fig. 7, pairwise comparisons showed that TARGETING (0.85s) was significantly faster than the other four tasks: p<0.001 for all, DUALTARGETING (0.91s, 7.1% faster), DOCKING (0.93s, 9.4% faster), THROWING (0.93s, 9.4% faster) and REORIENTING (1.12s, 31.8% faster). Moreover, REORIENTING was found significantly slower than other tasks: p<0.001 for all. We found a significant effect of *ID* ($F_{2.1,29.7}$=254.54, p<0.001, $\eta_p^2$=0.95). Pairwise comparisons showed significant differences (p<0.01) between each ID except for 1.81&2 bits and 2.07&2.37 bits. We also found a significant interaction effect between *Task* and *ID* ($F_{20,280}$=3.00, p<0.001, $\eta_p^2$=0.18). REORIENTING was significantly slower than other tasks for all IDs (p<0.05), except THROWING at 2.66 bits (p=0.08). TARGETING was only significantly faster than DUALTARGETING and REORIENTING at 1.49 bits, and THROWING at 2.37 bits (p<0.01), as shown in Fig. 8. There were no significant differences among other conditions.

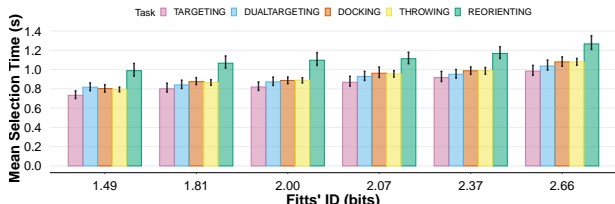

**Figure 8: Mean selection time of $T_{sel}$ by *ID* and *Task*. Error bars are shown with 95% confidence intervals.**

## 4.3 Motion Kinematics

In order to try to better understand where the difference for $T_{sel}$ between *Tasks* comes from, we analyzed the motion kinematics profile of participants' correct target selections. We once again kept all non-erroneous trials and removed the first block due to the aforementioned possible learning effect.

We computed velocity profiles based on the cursor's 3D position and corresponding timestamps. A time interval of each trial was normalized. Then, the corresponding velocities and normalized distances from the cursor to the *Primary Target* were interpolated to create a time-equidistant profile (every 2% of trial time). Next, for each task and participant, these values were aggregated by computing the average for each normalized time interval. These were subsequently averaged over all participants to produce a single normalized profile for each task (see Fig. 10).

In order to compare our results to Mandryk & Lough's work [38], we also used these profiles to compute the following motion kine-

matics measures: peak velocity (pkV), time to peak velocity (t2pkV), percent of time after peak velocity (afterpkV%), and selection velocity (sV: the velocity at the end of the $T_{sel}$) at *Primary Target* for each trial (see Fig. 9).

For each metric, we removed outliers if the observed value was more than three standard deviations from the mean. Given the non-normal distribution of dependent variables, we conducted an Aligned Rank Transform [61] for pkV, t2pkV, afterpkV% and sV on two IVs: *Task* and *ID*. Contrasts ART [20] with Bonferroni corrections was applied as the post-hoc analysis if a significant effect was found and effect sizes were reported as partial eta squared ($\eta_p^2$).

**Peak Velocity (Fig. 9 (A)):** We found a significant effect of *Task* on pkV ($F_{4,1306}$=76.67, p<0.001, $\eta_p^2$=0.19). Contrasts ART showed significant differences of pkV (p<0.001 for all) between each *Task* except between DUALTARGETING (M=51.52 cm/s) and DOCKING (51.52 cm/s). REORIENTING (49.1 cm/s) was 8.3% slower than TARGETING (53.2 cm/s), 4.8% slower than DUALTARGETING and DOCKING respectively, and 13.6% slower than THROWING (55.8 cm/s). We found a significant effect of *ID* on pkV ($F_{5,1306}$=608.60, p<0.001, $\eta_p^2$=0.70). Pairwise comparisons revealed significant differences (p< 0.001) between pairs of *ID* except for 1.49&2.00, 1.81&2.37 and 2.07&2.66 bits. We also found an interaction effect between *Task* and *ID* ($F_{20,1306}$=2.14, p<0.005, $\eta_p^2$=0.03). Contrasts ART also showed that there were no significant differences between 1.49&2.00, 1.81&2.37 and 2.07&2.66 bits for each task. In other words, no significant differences on pkV were revealed when target amplitudes were the same.

**Time to Peak Velocity (Fig. 9 (B)):** We found a significant effect of *Task* on t2pkV ($F_{4,1306}$=43.31, p<0.001, $\eta_p^2$=0.12). Pairwise comparisons showed that TARGETING (0.51 s) and DUALTARGETING (0.52 s) had significantly shorter t2pkV (p<0.001 for all) than the other three tasks. More specifically, they reached the peak velocity at least 3.8% earlier than DOCKING, THROWING and REORIENTING (0.54 s for all). We found a significant effect of *ID* ($F_{4,1306}$=27.56, p<0.001, $\eta_p^2$=0.10). Pairwise comparisons showed significant differences (p< 0.001) between each *ID* except for 1.49&2.00, 1.81&2.37, 2.07&2.37 and 2.37&2.66 bits. We did not find an interaction effect between *Task* and *ID*.

**Percent of Time after Peak Velocity (Fig. 9 (C)):** A significant effect of *Task* was revealed on afterpkV% ($F_{4,1306}$=182.49, p<0.001, $\eta_p^2$=0.36). The post-hoc analysis showed TARGETING (38.7%) had significantly smaller afterpkV% (p<0.001 for all) than the other four tasks while REORIENTING (49.5%) had significantly larger afterpkV% (p<0.001 for all). Participants spent 28% more time after peak velocity in REORIENTING compared to TARGETING and at least 20% longer than other three tasks, suggesting a longer deceleration phrase in REORIENTING. We found a significant effect of *ID* on afterpkV% ($F_{4,1306}$=200.95, p<0.001, $\eta_p^2$=0.43). Pairwise comparisons showed significant differences (p< 0.001) between each *ID* except for 2.00&2.07 bits. We did not find an interaction effect between *Task* and *ID*.

**Selection Velocity (Fig. 9 (D)):** We found a significant effect of *Task* on sV ($F_{4,1306}$=123.43, p<0.001, $\eta_p^2$=0.27) and the contrasts ART revealed that REORIENTING had significantly slower sV (7.70 cm/s, p<0.001 for all) and THROWING had significantly higher sV (11.87 cm/s, p<0.001 for all) than other tasks. Compared to other tasks, sV for REORIENTING was at least 25% slower and over 50% slower than THROWING. DUALTARGETING (10.59 cm/s) also had significantly higher sV than TARGETING (9.64 cm/s, p<0.01, 9.9% higher). We found a significant effect of *ID* on sV ($F_{5,1306}$=95.62, p<0.001, $\eta_p^2$=0.27). Pairwise comparisons showed significant differences (p< 0.01) between each *ID* except for 1.49&1.81, 1.49&2.07, 1.81&2.07, 2.00&2.37 and 2.37&2.66 bits. We also found an interaction effect between *Task* and *ID* ($F_{20,1306}$=2.51, p<0.001, $\eta_p^2$=0.04). Pairwise comparisons showed that there were no significant differences between 1.49&1.81, 1.49&2.07, 1.81&2.07, 2.00&2.37, 2.37&2.66 and 2.00&2.66 bits in each task. Interpreting these numbers in terms of target width and amplitude, these results argue that sV is significantly impacted by target width.

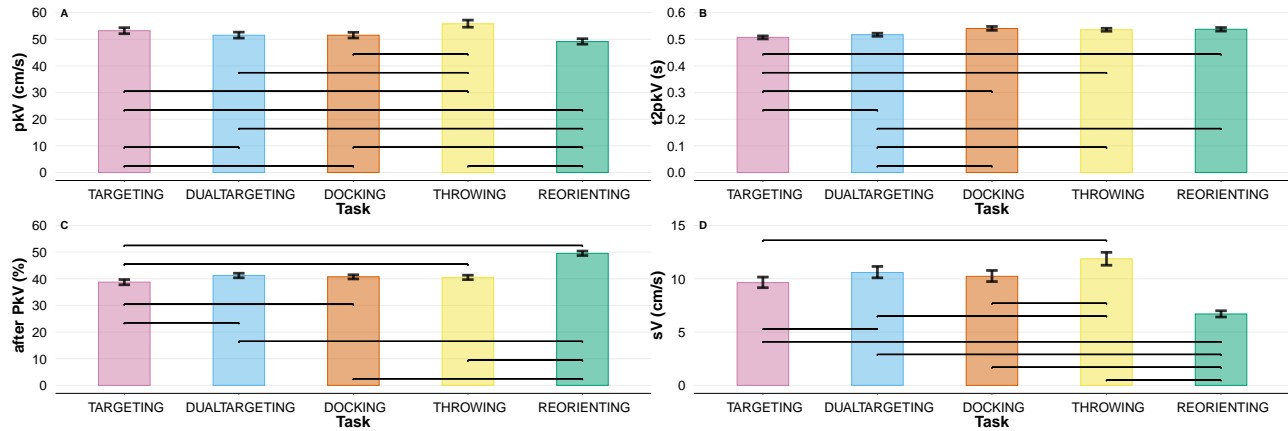

**Figure 9: Mean pkV (A), t2pkV (B), afterpkV% (C) and sV (D) of $T_{sel}$ for *Task*. Error bars are shown with 95% confidence intervals. The statistic significances evaluated by ART are connected with lines (p<0.01).**

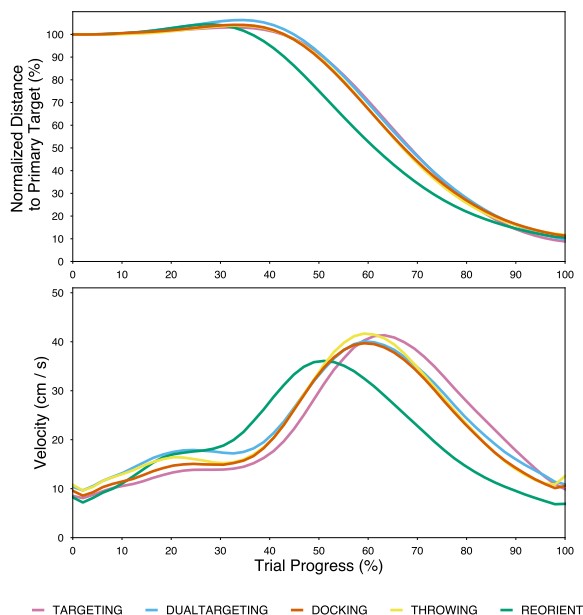

**Figure 10: Velocity profile and normalized distance percentage to *Primary Target* in $T_{sel}$. Trial progress refers to the normalized time interval of a trial.**

## 4.4 Fitts' Law

Given our research question, i.e. whether *Task* users will perform on an object impacts the time taken to acquire the object, Fitts' law modelling and throughput analysis were applied only to the selection of *Primary Target*, i.e., only to $T_{sel}$ in Figure 2 but not to the subsequent manipulation task. To perform our analysis, we computed the effective target width ($W_e$) for each target width by multiplying the standard deviation by 4.133 [37,41] and used $W_e$ to calculate the effective ID, $ID_e$ accordingly. As a result, the difference between $W$ and $W_e$ ranges from 0 cm to 0.6 cm and the corresponding difference between $ID$ and $ID_e$ ranges from 0 bits to 0.12 bits, as shown in Table 1.

The classical Fitts' Law was used for modelling [22, 36] due to two concerns: 1. for ID, as our targets are spheres, Fitts' law variants for targets in arbitrary shapes are not necessary. 2. as Triantafyllidis & Li [55] points out, no work has included all spatial factors in 3D space and a standard metric for 3D modelling is missing. Meanwhile,

the classical formula is still a common practice in VR [9, 54, 55].

| $A$(cm) | $W$(cm) | $W_e$(cm) | $ID$(bits) | $ID_e$(bits) |
|---|---|---|---|---|
| 9.0 | 3.0 | 3.2 | 2.00 | 1.93 |
| 9.0 | 5.0 | 4.4 | 1.49 | 1.61 |
| 12.5 | 3.0 | 3.0 | 2.37 | 2.37 |
| 12.5 | 5.0 | 4.6 | 1.81 | 1.89 |
| 16.0 | 3.0 | 3.0 | 2.66 | 2.66 |
| 16.0 | 5.0 | 4.8 | 2.07 | 2.12 |

**Table 1: Effective target width ($W_e$) and effective ID ($ID_e$) are calculated for each pair of target amplitude (A) and target width (W).**

To compensate for the non-normal distribution of selection time in $T_{sel}$, we adopted a common practice [14, 52] where we first aggregated the mean for each effective Index of Difficulty ($ID_e$) per *Block* per *Task* and then aggregated the median for each $ID_e$ and *Task*. $ID_e$ ranged from 1.61 to 2.66 bits and the aggregated median time of all tasks correlate with $ID_e$ positively ($R^2 \geq 0.96$), as shown in Fig. 11. When looking at the coefficients of these linear regression models in Table 2, we noticed that slope values $b$ were relatively similar across *Task* but REORIENTING has a higher y-intercept value $a$ than the other four tasks. We also report throughput scores in Table 2. Unsurprisingly, TARGETING had the largest throughput, REORIENTING had the lowest throughput, and the throughputs of the other three tasks were similar, a result consistent with $T_{sel}$ values.

| Task | a | b | TP |
|---|---|---|---|
| TARGETING | 0.36 [0.28 0.43] | 0.23 [0.20 0.28] | 2.47 [2.34 2.61] |
| DUALTARGETING | 0.44 [0.32 0.57] | 0.22 [0.16 0.28] | 2.29 [2.15 2.44] |
| DOCKING | 0.39 [0.30 0.49] | 0.25 [0.21 0.30] | 2.25 [2.13 2.38] |
| THROWING | 0.37 [0.27 0.48] | 0.26 [0.21 0.31] | 2.26 [2.13 2.38] |
| REORIENTING | 0.54 [0.38 0.70] | 0.27 [0.20 0.35] | 1.87 [1.76 1.99] |

**Table 2: Modelling results of Fitts' Law ($MT = a + b \cdot ID_e$) and throughput values $TP = ID_e/MT$: estimates (95% CI)**

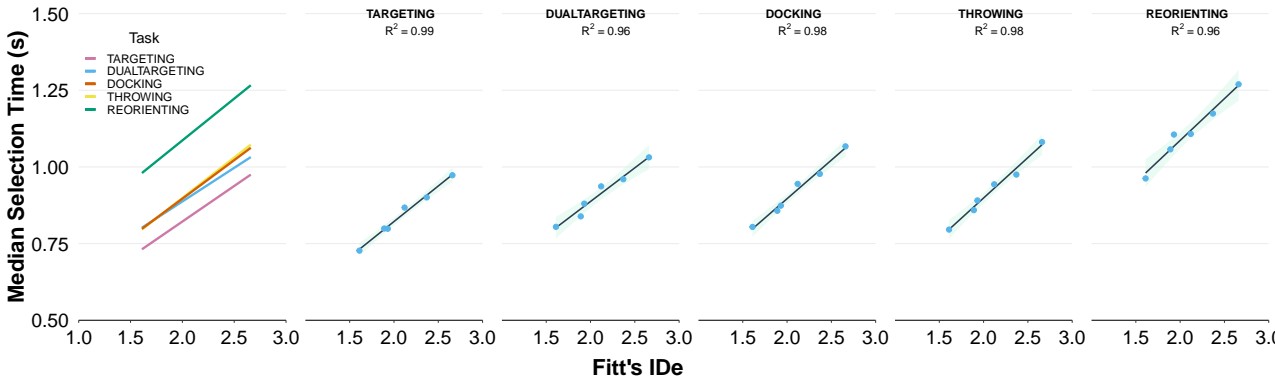

Figure 11: Median Selection Time as a function of Fitts' $ID_e$ per task, with corresponding $R^2$ and 95% confidence interval (light-green area).

## 5 GENERAL DISCUSSION

The goal of this paper is to explore the impact of intended use of a target, i.e. the *Task* performed on the target, on the time required to select that target, i.e. $T_{sel}$, before performing the manipulation. We examined five tasks: classical TARGETING, DUALTARGETING, DOCKING, THROWING, and REORIENTING. Our hypothesis is that there is an impact, i.e. that the selection that precedes target manipulation is impacted by the specific manipulation *Task* that we perform on the selected target.

Our results support this hypothesis. While we did not find a significant effect on error rate, we did find significant differences in the time taken to select the target. In particular, we found that selection preceding classical TARGETING took significantly less time than selections preceding all other *Task*s we tested, and that selection preceding a REORIENTING *Task* took significantly longer than selections preceding all other tasks. More specifically, with the interaction effect between *Task* and *ID*, TARGETING was significantly faster than some *Task*s at certain *ID*s and REORIENTING was significantly slower than all other *Task*s at all *ID*s except for THROWING at 2.66 bits. Target selection preceding DUALTARGETING, DOCKING and THROWING did not differ significantly in the time taken. While differences of selection time between those four tasks and TARGETING was small (within $0.27s$), they had been already more than 7% slower, with REORIENTING over 30% slower than TARGETING. Given that there are significant differences for TARGETING and REORIENTING, we reject our null hypothesis (no impact of intended use) and claim evidence for the impact of intended use on target selection.

To understand where and how this difference occurred, we analyzed movement time during the selection task. First, we created kinematic profiles of distance and speed during target selection. A visual inspection of Figure 10 shows that REORIENTING has a lower peak speed and reaches peak speed earlier in movement with a corresponding increase in the length of time spent decelerating. Further kinematic analysis supports this visual analysis; in Figure 9, REORIENTING has the lowest peak speed (at least 4.8% lower) and the longest deceleration phase (more than 20% greater) than other *Task*s by a significant margin. It implied that participants planned their movement and did not rush during the selection for REORIENTING. Besides, the Fitts' law modelling confirmed this by showing that the reaction time for REORIENTING was higher and the throughput value of REORIENTING was smaller than other tasks.

The kinematic analysis does, however, present some additional observations that merit future investigation. For example, REORIENTING results in the lowest peak speed; THROWING results in the highest peak speed in the preceding selection movement, which is followed by peak speed for selection preceding TARGETING. DUALTARGETING and DOCKING do not result in significant differences in peak speed in the preceding selection kinematics. Effect

sizes are not small for various measures of kinematics highlighted in Sect. 4.3. Based on typical interpretations of effect size, $\eta^2 > 0.14$, i.e. large effect size[1], for all differences except time to peak speed ($t2pkV$), where the effect size maps to medium effect. The absence of additional significant differences in selection time between DUALTARGETING, DOCKING and THROWING does not imply that differences in selection prior to these *Task*s do not exist; it simply implies that we measured no significant differences in selection time. Future work is planned to probe these effects in more depth.

It is interesting to contrast our results in Mandryk's and Lough's results in 2D [38]. Mandryk's and Lough's four intended uses (*Task*s) correspond to TARGETING, DUALTARGETING, DOCKING, and THROWING in our experiment. They found that TARGETING and DUALTARGETING resulted in selection times ($T_{sel}$) that differed significantly from DOCKING's and THROWING's selection times, but did not observe differences between TARGETING and DUALTARGETING. In contrast, while we, too, found that TARGETING resulted in shortest preceding selection times, DUALTARGETING resulted in $T_{sel}$ more similar to DOCKING's and THROWING's $T_{sel}$. In terms of overall differences, we and Mandryk and Lough find differences in $T_{sel}$ under 10% in overall magnitude for these selections (9.4% in ours and 8.8% estimated in theirs). Our REORIENTING is unique to our study, and resulted in the most significant differences in preceding selection time; REORIENTING resulted in target selection times more than 30% longer than TARGETING.

Given our results that the intended use of a target impacts the time taken to select a target, the question that follows from this is what these results mean. To address this question, we point, again, to the analysis of 20 years of VR-based by Bergström et al. [10]. As noted earlier in our review of this work, Bergström et al. highlight that, as dependent variables, selection task studies measure selection time for the selection task and manipulation task studies measure the time participants take to manipulate an object. Success in selection-based research is measured by shortening selection times or reducing errors, or both; success in manipulation is similarly based on increased throughput for the manipulation task. The assumption that underlies these success metrics is, *ipso facto*, that each individual user action can be optimized in isolation from other tasks in interfaces, i.e., that manipulation does not impinge upon preceding selection, but our results argue that this assumption cannot be made. If different manipulations impact the time taken for a preceding selection, then measuring only the manipulation time may over-estimate (or under-estimate) the benefits of a novel interaction technique.

It is also true that we only measure retrospective impact (i.e. the impact of future intended use on preceding selection), but prospective impacts are also possible. It is hypothetically possible that a pointing facilitation technique in VR, e.g. an area cursor, might

---

[1] https://www.spss-tutorials.com/effect-size/

impact a user's ability to perform a task on a target, e.g. reorientation of the target acquired via the area cursor. While exploring prospective impacts is one area of future work, it also highlights a more general implication for system design. Specifically, when we have new interaction techniques (i.e. new manipulations) or new pointing facilitation techniques (i.e. new selection techniques), incorporating them into realistic systems requires thinking not just about the individual action that they optimize but also about their place within and more general impacts on the overall task flow of the user.

## 5.1 Applicable Scenarios

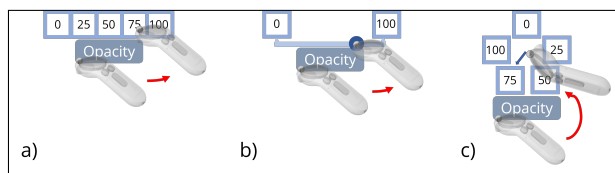

**Figure 12: Menu selection for changing target opacity: a) DUAL-TARGETING, b) DOCKING, c) REORIENTING.**

Our results can be framed into concrete applications and interface design in VR. One classical example is menu selection [21, 40]. Considering Fig. 12, our results suggest that interactions that take advantage of DUALTARGETING and DOCKING may result in similar selection time prior to the manipulation sub-task and differ from each other based on the design of manipulation techniques. In contrast, techniques that leverage REORIENTING already take longer selection time prior to the manipulation. Similarly, in data visualization tasks in virtual environments [12], rotational manipulations of a dataset may introduce additional costs if the target acquisition prior to the rotational manipulation is slowed.

## 5.2 Limitations

One highlighted limitation is that the range of IDs (1.49 to 2.66 bits) in this experiment is fairly small for a Fitts' Law design. This is because we restrict our current experiment to arms' length interactions with a controller on static targets, so target widths and distances are constrained for a reachability concern in VR. While these IDs are commonly used in a arms' length interactions in VR environment, they are low compared to desktop interfaces [38], touch-based interfaces [24], distant interactions in VR [7, 9, 14, 15], contexts where movement amplitudes can increase due to the greater distance of targets from the user. This explains why the throughput values in Table 2 are relatively lower, compared to throughput scores in other VR studies [9, 54]. However, incorporating more distant interactions adds additional complexity to the selection action because direction (e.g. targeting via a ray) and depth are often controlled differently during distant interaction [7, 9]. Future work could assess the findings for a wider range of IDs.

While we contrast five different *Task*s (the intended use) in our study, there exist more complex manipulations in the virtual environments. Furthermore, we do not consider objects' surface characteristics, perceived weight, and perceived fragility [30, 47] as objects in our study have similar surface, weight and fragility. We also do no consider bimanual interactions [27] nor co-articulated actions (e.g. 6-dof reorienting, i.e. a docking task that requires both rotation and translation) [29, 49]. As noted in the experiment design, we did not control the *Primary Direction* and *Secondary Direction*. Machuca and Stuerzlinger [8] found that target acquisition in virtual environments was slower and had less throughput along the depth-axis, i.e. Forward and Backward in our experiment, than lateral directions. Further research can explore how directions can impact the

acquisition for various intended uses, particularly for co-articulated actions.

Finally, it is noted that our experiment was conducted remotely in participants' homes. While an in-home environment increases the external validity of our study, it cannot be as controlled as a laboratory one, whose setup and control can assume to be optimized for the experiment. Space in homes may be constrained, and households may present interruptions during the experiment. To limit this as a factor, we note that participants were provided with detailed instructions, from both pre-recorded videos and experiment systems. A researcher was also present via video conferencing tools during the experiment, allowing the environment to be monitored for confounds. We also note that our within-subjects design partially controls for confounds by ensuring that the environment is similar across each task for a participant.

## 5.3 Future Work

The results of our intended use study present interesting avenues of future research into interaction in virtual reality environments. As one example, a novel interaction technique might not result in higher throughput during the interaction, but it is possible that it might speed the selection that precedes the interaction. In this case, considering both the selection and manipulation of a target as a unified task allows us to identify potentially beneficial novel interactions that might have been ignored if the only metric for success is throughput for the manipulation.

Another possible area of inquiry given differences in kinematics noted in Sect. 4.3 is that a more careful analysis of movement during selection might allow the system to infer what a user intends to do with a target prior to acquiring the target. This, in turn, could allow us to develop interaction techniques that leverage this inference. Reorientation appears a good initial candidate to identify given the deviation in the selection kinematics shown in Sect. 4.3.

## 6 CONCLUSION

A significant body of research in virtual reality exists that explores both selection and manipulations techniques to speed performance, and this past research typically assumes that these actions – the selection and the subsequent manipulation – can be independently optimized [10]. Based upon past work in 2D environments, in this paper we question this assumption. We examine the impact of five common virtual reality manipulations (TARGETING, DUALTARGETING, DOCKING, THROWING, and REORIENTING) on the time taken to select a target prior to performing the manipulation. We identify the existence of an effect of subsequent use of a target on the act of selecting the target. Specifically, we find that TARGETING had the shortest selection time, REORIENTING had the longest, and the other three intended uses we evaluate result in acquisition times between these two extremes. We synthesize these results and highlight their implications for research and design in VR environments.

### ACKNOWLEDGMENTS

We would like to thank all participants for their help with the remote studies, and reviewers for their valuable feedback. This research received ethics clearance from the Office of Research Ethics, University of Waterloo, and this work was made possible by the LAI Réapp. This paper was published in memory of Edward Lank, whose contribution to science will not be forgotten.

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
