# OpenReview forum: "Exploring the Effects of Intended Use on Targeting in Virtual Reality"
_graphicsinterface.org/Graphics_Interface/2023/Conference — GI 2023_

### Official Review · Reviewer_4jyH · 2023-01-12
**Paper review**

**Rating:** 9
**Confidence:** 4

**Review:**

This paper presents a study investigating the impact of the intended use on target acquisition in virtual reality. The initial hypothesis is that the intended use of the target (i.e. what the user intends to do with the target, such as select, drag, scale, rotate, etc.) can have an influence on target acquisition performance. A such, authors propose to consider such intention as an additional factors, beyond the traditional target distance and width that are considered in Fitt's Law.

The paper explores this question for the particular task of proximal target acquisition using a virtual hand metaphor, followed by five object manipulations (targeting, dual targeting, throwing, docking and reorienting). The results show that target acquisition is fastest when targeting and slowest when reorienting. Authors further analyse the movement profiles to differences in various movement properties such as peak speed or selection speed among others.

Review:

The paper is well written and the work is properly motivated. I find it an interesting idea to explore the intended use of the targeting in VR.

Novelty: While this hypothesis has already been tested on 2D interfaces, the question is still unanswered for 3D interfaces and in VR.

Moving from 2D to 3D is not trivial, as underlined by authors, as there are many differences: interaction devices, types of pointing (absolute, relative), depth perception, etc.. Hence addressing a particular task such as proximal target acquisition with a virtual hand (i.e. with a 1-to-1 mapping of physical controller movement to virtual controller movement) seems reasonable as a first step towards answering the initial question.

The paper is well illustrated, in particular I find the timeline of the trial to clearly present the experimental task, as well as the illustrations of the interaction and the graphs.

Design choices, such as the shape of the virtual controller, the directions of the targets, are well motivated. Overall, the experimental design is sound well thought and the study properly conducted. The experiment description is sufficiently detailed to facilitate the replicability of the work.

Results: the results are well analyzed and presented. The analysis of the motion kinematics allows to get mode depth in the results rather than just analyzing the time performance. My only comment is that the interaction between  ID and Task could be better explained, e.g. is Targeting still faster for all IDs ?

Finally, the discussion properly summarizes and discuss the results with respect to previous works, both on 2D pointing and VR interaction. I appreciated how authors provided an illustration of an application of their results, which is not always the case in pointing studies.

The paper acknowledges some limitations, such as the distant study, which I think do not question the validity of the results.

Overall, I think this paper presents an interesting and valid study, providing novel results compared to the related work. For all these reasons I advocate for accepting this paper to GI 2023.

---

### Official Review · Reviewer_eqq5 · 2023-01-13
**Very solid contribution, will be of wide interest, easy accept**

**Rating:** 9
**Confidence:** 5

**Review:**

This paper presents a study of the effects on targeting behaviour and performance for different task types in VR. Interesting work has previously established that targeting performance and kinematics change based on the task that the targeting action is part of (e.g., people target things more slowly when their task is to reorient an object then when it is to select it).

While this work has been established and replicated in the context of desktop pointing with a mouse, it has not been tested or evaluated in VR. While the contribution space might be considered a little narrow, it is my firm belief that this work will be widely used and cited with the increased amount of work in VR and MR that is occurring within HCI.

Additionally, the work is an exemplary execution of a "Fitts' " study. It does an excellent job of operationalizing the tasks and study procedure, and provides the analysis that would be expected. The work has interesting findings that are while in line with previous work with desktop pointing provides interesting considerations and discussion points for the VR context.

One critique I have for the authors that I think should be acknowledged or otherwise addressed by the authors:
The IDs used were very similar. For example, McKenzie typically uses a much wider range of IDs. Considerhttps://www.researchgate.net/publication/273742479_A_Note_on_the_Validity_of_the_Shannon_Formulation_for_Fitts%27_Index_of_Difficulty where data is presented and IDs are 1 to 7. This is a very large range, but the IDs presented in this paper are all within 1.2 bits, which are not very different. The cited work by Mandryk and Lough used IDs that span over 3 bits.

Overall, however, I don't believe this concern would meaningfully change the results given that the results seem to hold with previous work. And, so we have a very solid contribution that would make a good contribution to the literature.

---

### Official Review · Reviewer_beFU · 2023-01-17
**Valid study with relevant findings.**

**Rating:** 7
**Confidence:** 5

**Review:**

This paper looks at the effects of following activities on target acquisition performance. This work follows similar research on desktop interfaces where effects have been shown - the main contribution is extending these findings into VR. The authors run a Fitts' Law type study to compare the effects of five follow-up task conditions. The results show some differences in selection time, and further analysis reveals differences in peak velocity and timing during the pointing task.

The topic is applicable to current researchers and designers, and the results provide value. The study overall is well designed for answering the proposed research question and and the analyses are sound. My main critique is that the reporting is heavily focused on finding significant differences. I would be more interested in hearing about the differences between conditions and the implications of this for design. Some of the differences between means are quite small in this case, so it's not clear there are strong implications from these.

As acknowledged in the paper, the direction variables were randomised, so may have introduced some confounds into the results. It would be nice if the study design were able to eliminate potential confounds here, which may provide more power to the results.

A couple of further minor issues:
- It's good practice to avoid the combination of red and green cues in such studies to accomodate participants with red green colour blindness. This is the most common form of colourblindness, affecting about 1 in 12 males.
- The range of IDs is fairly small for a Fitts' Law design. However, this does not have a strong impact on the findings, as the main results were not in the throughput analysis.
- I suggest to avoid using citations in place of author names. This is a poor practice that causes distraction and makes the paper less readable.
- The authors may also be interested in this related paper:
Mayra Donaji Barrera Machuca and Wolfgang Stuerzlinger. 2019. The Effect of Stereo Display Deficiencies on Virtual Hand Pointing. In Proceedings of the 2019 CHI Conference on Human Factors in Computing Systems (CHI '19). Association for Computing Machinery, New York, NY, USA, Paper 207, 1–14. https://doi.org/10.1145/3290605.3300437

Despite a few limitations, the study is overall well done and provides some findings worth sharing with the research community.

---

### Meta-Review · Area_Chair_erAG · 2023-01-17

**Recommendation:** strong accept
**Confidence:** 5

**Metareview:**

The paper has received three reviews from reviewers who are knowledgeable of the topic and felt confident in their reviews. All three recommended acceptance and felt the work was well executed. It is easy to recommend acceptance.

The reviewers request three changes for the final paper:

1) please consider how to address the critique from R3, “ I would be more interested in hearing about the differences between conditions and the implications of this for design. Some of the differences between means are quite small in this case, so it's not clear there are strong implications from these.” Please address this critique.

2) please acknowledge and discuss the limited IDs used in the study.

3) please provide a more detailed explanation on the interaction between ID and Task, e.g. is Targeting still faster for all IDs ?

I believe these changes to be quite minor and all reviewers were confident in this papers contribution.

Congratulations!